# Remember what you did so you know what to do next

**Manuel R. Ciosici, Alex Hedges, Yash Kankanampati, Justin Martin,**
**Marjorie Freedman**, **Ralph Weischedel**

manuelc@isi.edu, mrf@isi.edu, weisched@isi.edu

Information Sciences Institute, University of Southern California

## Abstract

We explore using a moderately sized large language model (GPT-J 6B parameters) to create a plan for a simulated robot to achieve 30 classes of goals in ScienceWorld, a text game simulator for elementary science experiments. Previously published empirical work claimed that large language models (LLMs) are a poor fit (Wang et al., 2022) compared to reinforcement learning. Using the Markov assumption (a single previous step), the LLM outperforms the reinforcement learning-based approach by a factor of 1.4. When we fill the LLM's input buffer with as many prior steps as possible, improvement rises to 3.5x. Even when training on only 6.5% of the training data, we observe a 2.2x improvement over the reinforcement-learning-based approach. Our experiments show that performance varies widely across the 30 classes of actions, indicating that averaging over tasks can hide significant performance issues.

In work contemporaneous with ours, Lin et al. (2023) demonstrated a two-part approach (SwiftSage) that uses a small LLM (T5-large) complemented by OpenAI's massive LLMs to achieve outstanding results in ScienceWorld. Our 6-B parameter, single-stage GPT-J matches the performance of SwiftSage's two-stage architecture when it incorporates GPT-3.5 turbo which has 29-times more parameters than GPT-J.

## 1 Introduction

Our research interest is in using a modest-sized, self-hosted large language model (LLM) for planning so that a robot can execute the plan(s) to achieve goals. Google's *InnerMonologue* project (Huang et al., 2023) uses PALM (540B parameters) and a real robot; Lin et al. (2023) use a small LLM (T5-large) but for many decisions relies on the massive model GPT-4 (OpenAI, 2023) to achieve goals in ScinceWorld. Our study uses GPT-J (6B parameters, Wang (2021)) in an open-source, simulated environment, ScienceWorld (SW,

Wang et al. (2022)). We selected ScienceWorld for this study since it (1) supports several types of primitive actions, many types of entities, and 30 classes of tasks representing significantly differing challenges; (2) provides a simulator for the environment, including natural state changes (e.g., ice melting, a butterfly emerging from a chrysalis); (3) may benefit from applying common sense knowledge; (4) might be improved by external knowledge from a knowledge graph or from text; and (5) had a substantial amount of training data so that we could explore the effect of training set size. Our empirical results contrast with those of Wang et al. (2022), who evaluated five architectures via individually training 30 different models, one for each of the 30 classes of tasks. Those models chose the next step given the current text feedback from the simulator and the previous action (i.e., Markov assumption). By contrast, we train a single GPT-J model to cover all 30 classes of tasks and explore filling the LLM's input buffer with as much game history as fits. A contemporaneous study (Lin et al., 2023) uses two models: T5-large makes routine decisions; when needed for some decisions, the architecture calls GPT-4. Though performance relying on GPT-4 exceeds GPT-J, when Lin et al. (2023) used GPT-3.5 Turbo, rather than GPT-4, performance is comparable to our results with GPT-J which uses 29-fold fewer parameters than GPT-3.5 Turbo.

What we have learned is:

1. By including a sequence of previous actions, not just the most recent one, the average score triples compared to DRRN.
2. Performance of the LLM model degrades gracefully with much less training data. With only 6.5% of the training data, the score doubles compared to DRRN.
3. If one uses all the training data provided, GPT-J learns enough that adding a pre-condition system text does not help (i.e., overall performance stays the same or drops slightly).

4. Despite previous findings that LLMs perform poorly in ScienceWorld, we find they outperform the DRRN by a factor of 3.5.

5. The 6B parameter GPT-J can match the performance of complex dual-LLM architectures that rely on LLMs 29 times larger than GPT-J.

## 2 Related Work

**Reinforcement Learning (RL)**. Jansen (2022) systematically surveyed the available text game environments, RL techniques, and some of the field's challenges, such as low-complexity text environments and the limitations of classic planning techniques such as PDDL (Ghallab et al., 1998) when it comes to text game environments. Two of the surveyed RL methods (DRRN and CALM) achieved the highest scores in the original ScienceWorld evaluation (Wang et al., 2022). DRRN (He et al., 2016) is one of the most successful RL methods for text games. When playing ScienceWorld, DRRN requires assistance from the game in the form of a list of valid actions to choose, as ScienceWorld's action space is too large for the model (Wang et al., 2022). CALM (Yao et al., 2020) attempts to address DRRN's need for a list of valid actions by using a language model (GPT-2) to generate a set of actions for its RL component to choose from. Despite succeeding in some text games, RL struggles to play games with large action spaces or those that require background knowledge not learnable from the game.

**Language models** have recently surpassed RL methods in traditional RL benchmarks, including playing Atari games (Chen et al., 2021; Janner et al., 2021). Kim et al. (2022) used DistilBERT (Sanh et al., 2020) for its common sense to improve on DRRN in Zork1, Zork3, and 12 other text games that pre-date ScienceWorld. Singh et al. (2021) explored building on BART (Lewis et al., 2020) as a world model for playing Zork1 and eight other text games that pre-date ScienceWorld. Most recently, Lin et al. (2023) employed T5 (Raffel et al., 2020) for an initial plan to achieve ScienceWorld goals and GPT-4 (OpenAI, 2023) to re-plan when the T5 model was challenged.

Language models generally approach the problem as a sequence modeling task, performing the equivalent of *offline reinforcement learning* (Levine et al., 2020). LLMs can also store large amounts of world knowledge, giving them a potential advantage over pure RL methods in environments that require prior knowledge, like ScienceWorld. Some recent work has shown how extremely large language models (over 100B parameters) can plan in simulated and real-world environments (Huang et al., 2022; Ahn et al., 2022; Huang et al., 2023). *SayCan* (Ahn et al., 2022) and *Inner Monologue* (Huang et al., 2023) use a huge language model (e.g., a 540B parameter PaLM, Chowdhery et al. (2022)) to create a plan for a robot to convert into actions. In contrast to InnerMonologue, the system in this paper uses a language model that is 90 times smaller (6B parameter GPT-J, Wang (2021)), which both plans and turns its plan into low-level actions, though in a simplified, simulated environment.

## 3 The task

### 3.1 ScienceWorld Description

ScienceWorld (Wang et al., 2022) is a multiroom text game that tests scientific reasoning abilities at the level of a standard elementary school science curriculum. It contains 30 different classes of tasks, each with many variations that challenge the player to adapt to environmental changes. The variations challenge players to demonstrate working knowledge of scientific concepts and experiments rather than the declarative knowledge usually demonstrated by question-answering LLMs. For example, some task variations prevent memorization (e.g., by randomizing object locations). Others require a different experimental path by changing the nature of the task (e.g., melting aluminum instead of ice) or by adding obstacles (e.g., a broken stove requires players to find another heat source). ScienceWorld also tests players' ability to generalize by including tasks where the target object is known and can, therefore be solved using prior knowledge (e.g., determining the boiling point of water) and tasks where the target object is unknown, and the player must perform a scientific experiment (e.g., determining the boiling point of an unknown liquid substance). There are 7 207 total task variations across the 30 tasks, split unevenly among the tasks with a minimum of 10 variations per task and a maximum of 1 386. The game allocates 50% of all variations for *training*, 25% for *development*, and 25% for *testing*.

ScienceWorld has over 1 000 possible actions, of which a few tens or hundreds are valid at any point. Players receive points when they achieve task-specific milestones and a score of 0 if they do

not perform the required task or perform it incorrectly, e.g., focusing on an animal when the task is to find a plant. Furthermore, the design of the *train/dev/test* split is such that players do not encounter test tasks during training. For example, if melting ice is a training task, test and dev include variations of the melting task that involve different materials (e.g., aluminum), requiring different temperatures, and even alternative heat sources (e.g., a forge instead of a stove). The *train/dev/test* setup makes the game difficult for traditional RL systems as it requires agents to generalize from the train set based on background world knowledge, which RL agents typically do not have.

## 3.2 Data generation

For any task variation, ScienceWorld can automatically generate one or more unique gold paths consisting of actions that will achieve the required goals. For training data, we generate up to 3 unique gold paths for variations in the train set, yielding 7 359 unique gameplays. From this training data set, we sample several smaller train sets. We create one training set containing only one unique gold path per task variation, yielding 3 589 unique gameplays. We also create a task-balanced training set by sampling a maximum of 18 gameplays per task from the half-sized training set[1], resulting in 480 unique gameplays, almost evenly distributed across the 30 ScienceWorld tasks but including only a small subset of each task's variations, thus challenging agents to generalize from much less varied data. All the training data we used is available at `https://github.com/isi-vista/science_world_data`.

## 4 Results

### 4.1 Training

We fine-tuned GPT-J on ScienceWorld games transcribed as a natural language dialog between an agent and the game. The agent issues actions; the game replies with the current observation but does not mention the score. This plain text sequence-modeling format differs from prior approaches to framing the RL problem as a sequence modeling task (see Appendix A for details). Formulating games as dialog transcripts allows us to use any autoregressive pretrained LLM and take advantage of the vast prior knowledge encoded in modern

transformer-based LLMs. Except for one experiment condition, we fill GPT-J's input buffer with as much prior history as possible.

We train GPT-J-based models for the following conditions: **All train Markov** uses the entire training data of 7 359 games, but only gives GPT-J the prior action and game observation (i.e., uses a Markov assumption, like the agents tested by Wang et al.); **All train** uses the entire training data set and fills GPT-J's input buffer; **No variations** uses roughly half of the training data (as described in the previous section) and fills GPT-J's input buffer. Finally, **Up to 18 games** is trained only on the small subset of 18 task variations per task (480 games, approximately 6.5% of the entire training data) but fills GPT-J's input buffer. We include a complete description of the training details and hyper-parameters in Appendix B.

To evaluate potential improvements to the LLM from external components, we designed a preconditions checker to assist the LLM. *The preconditions system* parses ScienceWorld's text descriptions and keeps track of the state of doors and drawers (open or closed). If GPT-J attempts to reopen an object already in an open state, the preconditions system intercepts GPT-J's output preventing it from reaching ScienceWorld. The preconditions system then replies to GPT-J pretending to be the game and to have performed the requested action. By preventing redundant actions from reaching the game, the preconditions system prevents GPT-J from wasting game turns on needless close or open actions. The preconditions system aims to add programmatic support for common sense and alleviate LLM tendencies to occasionally emit redundant actions.

### 4.2 Evaluation

Unlike Wang et al. (2022), who evaluated using a sample of 300 *dev* games (16.5% of *dev*), or Lin et al. (2023) who evaluated on a sample of only 270 *test* games, we evaluate over the full *test* set of 1 819 games. The prior literature is also inconsistent about computing scores. As in this work, Wang et al. (2022) assume that failed games have a score of 0 while Lin et al. (2023) use the score just prior to the agent failing. We include a detailed comparison of our testing setup with that of Wang et al. (2022) and Lin et al. (2023) in Appendix C. Like Wang et al. (2022), we report results after evaluations with an environment limit of 100 actions/steps, meaning that during the evaluation,

---

[1]For tasks with fewer than 18 variations, we take as many gold paths are there are task variations.

| | | Train | Score | Std. Dev. | Improv. | Games | | Actions (% of total) | | | | | |
|---|---|---|---|---|---|---|---|---|---|---|---|---|---|
| | | | | | | Won | Lost | Valid | AVs | IOs | IS | RAs | Other |
| 1 | **DRRN** | N/A | 17.95 | | 1.0x | | | | | | | | |
| | **GPT-J** | | | | | | | | | | | | |
| 2 | All train Markov | 7 359 | 24.74 | 1.05 | 1.4x | 117 | 74 | 61.13 | 3.51 | 31.14 | 2.43 | 1.77 | 0.02 |
| 3 | All train | 7 359 | 62.57 | 4.32 | **3.5x** | 1 012 | 383 | **90.51** | 0.06 | 4.68 | 2.51 | 2.07 | 0.17 |
| 4 | No variations | 3 589 | **63.35** | 6.94 | **3.5x** | **1 037** | 372 | 90.39 | 0.07 | 4.28 | 3.08 | 2.00 | 0.19 |
| 5 | Up to 18 games | 480 | 39.78 | 2.35 | 2.2x | 479 | 682 | 83.59 | 0.39 | 11.46 | 2.81 | 1.46 | 0.30 |

Table 1: ScienceWorld scores over the 1 819 games that comprise the *test* set. The train column shows the number of *train* games available during training. Improv. = relative improvement over DRRN. AVs = Affordance Violations; IOs = Invalid Objects; IS = Invalid Syntax; RAs = Redundant Actions. Note that the sum of won and lost games does not equal the total number of *test* games; as in many games, agents obtain a score that is neither 0 (lose) nor 100 (win).

games end if an agent fails, wins, or reaches the action/step limit. We discuss the effect of the 100 action limit on evaluation in Appendix E.

Out of the box, ScienceWorld only reports a player's score, no other performance metrics. To better understand a model's behavior, we analyzed our game transcripts to count the numbers of games lost and won (0 or 100 points) and to classify each action emitted by GPT-J as either valid or one of five categories of invalid actions: Affordance Violations (AVs, e.g., the agent tried to pour a chair), Invalid Objects (IOs, i.e., the agent tries to interact with a non-existent, hallucinated object), Invalid Syntax (IS, i.e., the action is valid English, but is not a valid ScienceWorld command), Redundant Actions (RAs, e.g., the agent tries to open an already open door), and Other. We present our results in Table 1, where each score is the mean over five training runs with different random seeds. All our GPT-J-based agents outperform the DRRN, the best-performing agent in the original ScienceWorld evaluation (Wang et al., 2022).

**All train Markov**, the GPT-J model trained on the entire training data using the Markov assumption (i.e., conditioning actions only on the prior action and the game feedback) outperforms DRRN by a factor of 1.4 (row 2). Only 61.13% of the actions emitted by this model are valid, and almost a third (31.14%) involve non-existing objects. Our result starkly contrasts those of Wang et al. (2022) who evaluated an LLM trained with the Markov assumption (a T5 architecture (Raffel et al., 2020) initialized with the Maccaw weights (Tafjord and Clark, 2021)) and found it performed poorly. Despite T5 being twice the size of the GPT-J in our experiments (11B parameters vs. 6B), it only ob-

tained 8 points on the entire test set, 3.1 times less than our Markov-instructed GPT-J (our weakest model). We postulate that our Markov-instructed GPT-J agent outperforms T5 due to our more natural formulation of the task and GPT-J's longer maximum input length (2 048 word pieces vs. 512).

**All train**, the model trained on the entire set of 7 359 games, and which conditioned its actions on as much history as possible (on average the previous 43.42 actions, see Appendix A), outperformed DRRN by a factor of 3.5 (row 3). The model won 1 012 games (55% of the 1 819 test games) and rarely emitted invalid actions (90.51% action validity).

Adjusting the training data such that each variation only appears with a single solution reduces the training data to slightly less than half. However, the resulting model (**No variations**, row 4) obtains a score almost one point higher than using all training (row 3). The model still wins, on average, 1 037 games (57%) with an action validity of 90.39%.

If we evaluate these two non-Markov models (*All train* training and *No variations* using the evaluation methodology of Lin et al. (2023), the scores change little (62.59 vs. 62.57 for *All train*; 61.24 vs. 63.35 for *No variations*). While these scores are far from the top performing SwiftSage system (*T5 + GPT-4*), they are similar to the 62.22 score obtained by the *T5 + GPT-3.5-turbo* SwiftSage system (Lin et al., 2023, Table 3). Thus, the single model 6B-parameter GPT-J model closes the gap to the approximately 29 times larger dual model which incorporated GPT-3.5-turbo.

Interestingly, even when only 480 games are available for training (a mere 6.5% of the training data), GPT-J learns to play ScienceWorld and

|  | RAs (%) | +P. RAs (%) | Score |
|---|---|---|---|
| All train Markov | 1.77 | 0.95 | +0.03 |
| All train | 2.07 | 0.99 | −0.06 |
| No variations | 2.00 | 0.74 | −0.14 |
| Up to 18 games | 1.46 | 0.67 | +0.07 |

Table 2: Percentage of Redundant Actions (RAs) emitted by our models before and after adding *the preconditions system* (+P) and the change in SW scores.

still achieves a 2.2x improvement over DRRN (**Up to 18 games**, row 5). This model wins far fewer games (26.3%), emits fewer valid actions (83.59%), and has a tendency to try to interact with hallucinated objects (11.46% of its actions involve non-existing objects). Despite the lower improvement over DRRN, this result indicates that GPT-J can generalize from even small amounts of training data. We hypothesize that the background knowledge GPT-J gained during its pre-training drives its ability to generalize from such little training data.

All our results have a standard deviation of only a few points, indicating good stability over the five training runs with different seeds. However, like Wang et al., we find that our models' performance varies greatly across the 30 tasks. We discuss the per-task performance in Appendix D.

We show the effect of the preconditions system in Table 2. The preconditions system almost halves the percentage of redundant actions passed from our models to ScienceWorld. However, the ScienceWorld score changes only a fraction of a point, inconsistently, up or down. The lack of change in the ScienceWorld score is not surprising. We evaluate all models with an environment step/actions limit of 100, meaning we can interpret the percentages in the *RAs* column as the average count of redundant actions per game. Even if the preconditions system prevented all redundant actions, that would only give each agent about two extra actions per game, not enough to influence the final score meaningfully. Despite the lack of increase in the ScienceWorld score, the preconditions system achieves the goal of removing non-common sensical actions from the LLM-game interaction.

## 5    Discussion and Conclusion

LLMs still exhibit non-common-sensical behaviors, such as emitting redundant actions or trying to interact with nonexistent objects. External components such as the precondition system shown in this work can help alleviate some of these behaviors. Other external components could assist LLMs with structured knowledge. Such knowledge could be quickly inspected and updated by humans. Our team has yet to succeed in significantly assisting the LLM using external knowledge sources. While we have managed to teach the LLM to interact with a Q&A system that could answer Is-A questions for the four classification tasks, such knowledge was already present in the LLM, as evidenced by the high scores on classification tasks in Appendix D.

Despite Wang et al. (2022)'s findings that LLMs perform poorly in ScienceWorld, we find that with a careful formulation of the prompt and access to prior interactions with the game, even a single reasonably sized LLM achieves a 3.5x improvement over DRRN. The 6B-parameter GPT-J models can match the performance of SwiftSage *T5 + GPT-3.5-turbo*, a more complex architecture that uses a model 29 times larger than GPT-J.

Even when learning from just a few hundred games (6.5% of the training data), GPT-J achieves a 2.2x improvement over DRRN. Despite the sizeable overall score improvement, the LLM performs poorly on some ScienceWorld tasks. The strong correlation between the performance of our GPT-J-based models suggests that some tasks are genuinely more difficult for the LLM.

## Limitations

This paper shows that an agent based on GPT-J, a Large Language Model (LLM), can perform elementary science experiments in a simulated text environment. Our GPT-J agent outperforms the state-of-the-art reinforcement learning models by a factor of 3.5 using a single model rather than DRRN's 30 task-specific models. While GPT-J's performance is impressive, readers should remember that Large Language Models like GPT-J have unforeseen gaps in knowledge and their behavior is often unpredictable. It is appealing to look at our experience with GPT-J and assume that one can fine-tune LLMs and then task them with operating machinery or robots in the real world. However, due to LLMs' unpredictable behavior, such an approach could result in material damage or even injure humans, animals, or the environment. This warning also applies to cases where LLMs are allowed to operate computer APIs which, despite their virtual nature, can have undesired effects in the real world. Should LLMs be allowed to oper-

ate APIs or real-world machinery, they should not be given complete control over the tools but operate within carefully chosen boundaries strictly enforced through software or through physical means.

When considering using LLMs to plan and execute in new domains, one should remember that the LLM used in this paper benefited not just from the opportunity to learn from example experiments but also from its background knowledge. By its nature, knowledge of elementary science is widely spread in texts on the web, such as those that comprised GPT-J's pre-training data. For a detailed presentation of GPT-J's pre-training corpus, we direct readers to Gao et al. (2020).

Finally, when interacting in ScienceWorld, LLMs have a text-only view of the environment. The environment is described via text with just the detail necessary for accomplishing the tasks, akin to having access to perfect Computer Vision and a system that can filter out trifling information, allowing the LLM to focus on planning. Access to perfect Computer Vision and relevance filters is a substantial limitation common to all approaches that operate in text-only environments. More research is necessary to understand how to teach LLMs to incorporate information from other modalities, such as sight and sound. Approaches such as PaLM-E (Driess et al., 2023) have shown that it is possible to integrate multiple modalities into Extremely Large Language Models. But, at 526B parameters, these models' compute and energy requirements seem to make them impractical for onboard processing of mobile robotics platforms.

## Ethics Statement

Environmental impact questions arise for any scientific work that involves Large Language Models (LLMs). Most of the energy consumption of LLMs occurs during pre-training (Patterson et al., 2021). This work relies on already pre-trained language models, which we only fine-tune. GPT-J, an LLM with 6B parameters, is small enough to be fine-tuned on a single modern compute node, contrasted with LLMs using hundreds of billions of parameters. For fine-tuning, we use nodes with 4x NVIDIA A6000 GPUs, and we further increase the training efficiency by offloading the optimizer from GPU memory via DeepSpeed (Ren et al., 2021; Rajbhandari et al., 2020), thus eliminating the need for the newest, most power-hungry GPUs.

## Acknowledgements

This material is based on research supported by DARPA under agreement number N66001-19-24032. The U.S. Government is authorized to reproduce and distribute reprints for Governmental purposes, notwithstanding any copyright notation thereon. The views and conclusions contained herein are those of the authors and should not be interpreted as necessarily representing the official policies or endorsements, either expressed or implied, of DARPA or the U.S. Government.

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

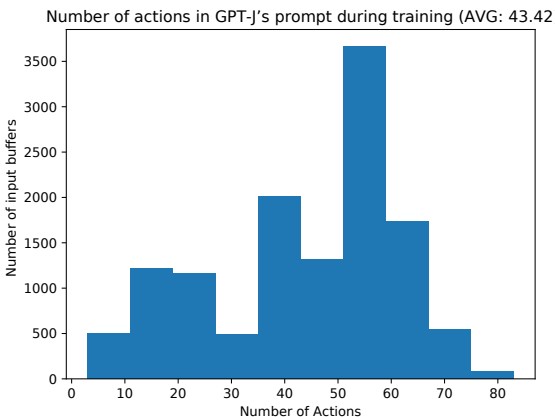

Figure 1: Histogram illustrating the number of actions that fit in GPT-J input for the *All train* training set.

## A    Sample ScienceWorld dialog

We fine-tune GPT-J on a formulation of Science-World games transcribed as a natural language dialog between an agent (A) and the game (G). The agent issues actions, and the game replies with the current observation. Unlike prior work that modeled RL algorithms using language models, we provide no information about the current score or score yet to be achieved. Listing 1 illustrates the dialog format. Training GPT-J becomes an autoregressive text modeling task over the dialog format. During inference, we fill GPT-J's input buffer with as much history as possible in the form of prior $(action, observation)$ tuples in the dialog format and expect GPT-J to generate an action row (i.e., the text after $A$ :). GPT-J's input buffer, limited to a maximum of 2048 word pieces, can accommodate, on average, 43 prior actions using the dialog format (Figure 1).

Our plain text, natural language sequence modeling format differs from prior approaches to framing Reinforcement Learning as a sequence modeling task. Instead of natural language input, Janner et al. (2021) use special tokens to represent a discretized version of the previous states and actions as input to a transformer decoder that predicts the next action. Notably, the transformer is a bespoke model, not a pre-trained Language Model. For The Decision Transformer (Chen et al., 2021), raw inputs are projected into a continuous space using a learned linear projection, and the projections become The Decision Transformer's input. Another difference between our simplified dialog modeling technique and The Decision Transformer is that our input

does not contain information about the game score of the yet unrealized score. This frees up space in the LLMs input buffer to allow for more prior history.

## B  Training details

We trained GPT-J with *bfloat16* representations using DeepSpeed stage 2 model parallelism over 4 NVIDIA A6000 GPUs. We used the AdamW optimizer with a weight decay of 0.01 and offloaded it to system memory via DeepSpeed (Ren et al., 2021; Rajbhandari et al., 2020). We train each model for two epochs over its corresponding train data and a batch of 16 (one input per batch, four gradient accumulation steps, 4 GPUs) and a learning rate of 1e-4.

## C  Evaluation setup

After training on each model's respective training data, we evaluated each agent on the task variations provided in ScienceWorld's test set. The evaluation required agents to play a total of 1 819 games split unevenly over the 30 tasks. The distribution of games to tasks appears in the *Games* column of Table 5.

By contrast, Wang et al. (2022) measured DRRN's performance against a random sample of the *dev* set every 500 steps during DRRN's training and reported the average score for the last 10% of training steps. For a fair comparison between DRRN and our models, we trained DRRN using the published code and parameters, evaluated it against the entire ScienceWorld *test* set, and reported DRRN's performance in Table 1.

Before we tested our trained DRRN model on the *test* set, we confirmed that our training was successful by performing the same evaluation as in the original ScienceWorld paper. In this evaluation, our trained DRRN obtained 18.92 points, similar to the published 17 points. Since these results are close, we conclude that our retrained DRRN model matches the capabilities of the DRRN model from the original ScienceWorld paper. For completeness, we also evaluated our trained DRRN model on the entire *dev* set, where it obtained 18.11 points compared to the 17.95 that it obtained when evaluated on the entire *test* set (i.e., the result we include in Table 1). The difference in performance between DRRN on *dev* and *test* is similar to what we observed for our GPT-J-based models.

One final thing to note is that there are several ways to compute mean performance in ScienceWorld. In Table 1, we report scores averaged over the 1 819 games in the *test* set. This mean gives equal weight to each game in the set. But, since variations are nonuniformly distributed over tasks, the contribution of each task to the final score is determined by the number task of variations in the *test* set (e.g., the *Changes of State (Boiling)* task only has 9 variations in *test*, while *Friction (known surfaces)* has 348 variations). Wang et al. (2022) computed the mean score by calculating the mean score per task and then the mean score over the 30 tasks. This method gives equal weight to each task while potentially obscuring failures in tasks with many variations. The difference between the two is similar to that between *micro-* and *macro-precision*, and both methods are valid. In Table 3, we compare the *micro-* and *macro-* averaged scores. The macro score is lower than the micro score for GPT-J-based models. Nonetheless, the best-performing GPT-J-based model still outperforms DRNN by a factor of 2.7.

|  | Micro Score | Improv. | Macro Score | Improv. |
|---|---|---|---|---|
| DRRN | 17.95 | 1.0x | 18.75 | 1.0x |
| All train Markov | 25.19 | 1.4x | 22.73 | 1.2x |
| All train | 62.57 | **3.5x** | **50.60** | **2.7x** |
| No variations | **63.35** | **3.5x** | 50.61 | **2.7x** |
| Up to 18 games | 39.78 | 2.2x | 34.29 | 1.8x |

Table 3: Comparison of *micro-* and *macro-* scores.

## D  Scores per Task

Table 1 shows that our GPT-J-based agent can achieve up to 3.5x the score of DRRN. Looking into the average scores separated by task provides further insight into the successes and shortcomings of the LLM approach to playing ScienceWorld. Table 5 shows results split by task and by task category. Each row in the table is based on five training and evaluation runs with different seeds. In each cell, we present the mean score and the standard deviation.

Like Wang et al. (2022), we find that performance varies considerably from task to task. Even for our best model, **All train**, the scores range from 0 points (on *Mixing paints (tertiary colors)*) to 100 points on several tasks, e.g., *Identify longest-lived animal*. However, it is not just the score that varies from task to task. We observe many variations in

Listing 1: An example transcript as used to train GPT-J.

```
Your task is to measure the melting point of chocolate, which is located around
    the kitchen. First, focus on the thermometer. Next, focus on the chocolate.
    If the melting point of chocolate is above -10.0 degrees celsius, focus on
    the blue box. If the melting point of chocolate is below -10.0 degrees
    celsius, focus on the orange box. The boxes are located around the kitchen.

A: look around
G: This room is called the hallway. In it, you see: the agent; a substance
    called air; a picture. You also see: A door to the art studio (that is
    closed); A door to the bedroom (that is closed); A door to the greenhouse (
    that is closed); A door to the kitchen (that is closed); A door to the
    living room (that is closed); A door to the workshop (that is closed)
A: inventory
G: In your inventory, you see: an orange
A: open door to kitchen
G: The door is now open.
A: go to kitchen
G: You move to the kitchen.
A: look around
G: This room is called the kitchen. In it, you see: the agent; a substance
    called air; a blue box (containing nothing); a chair. On the chair is:
    nothing.; a counter. On the counter is: a bowl (containing a red apple, a
    banana, an orange, a potato), a drawer.; a cupboard. The cupboard door is
    closed. ; a freezer. The freezer door is closed. ; a fridge. The fridge door
     is closed. ; a glass jar (containing a substance called sodium chloride); a
     lighter; a orange box (containing nothing); a oven, which is turned off.
    The oven door is closed. ; a painting; a sink, which is turned off. In the
    sink is: nothing.; a substance called soap; a stopwatch, which is
    deactivated. ; a stove, which is turned off. On the stove is: nothing.; a
    table. On the table is: a glass cup (containing nothing).; a thermometer,
    currently reading a temperature of 10 degrees celsius. You also see: A door
    to the bathroom (that is closed); A door to the hallway (that is open); A
    door to the outside (that is closed)
A: pick up thermometer
G: You move the thermometer to the inventory.
A: focus on thermometer in inventory
G: You focus on the thermometer.
A: look around
G: This room is called the kitchen. In it, you see: the agent; a substance
    called air; a blue box (containing nothing); a chair. On the chair is:
    nothing.; a counter. On the counter is: a bowl (containing a red apple, a
    banana, an orange, a potato), a drawer.; a cupboard. The cupboard door is
    closed. ; a freezer. The freezer door is closed. ; a fridge. The fridge door
     is closed. ; a glass jar (containing a substance called sodium chloride); a
     lighter; a orange box (containing nothing); a oven, which is turned off.
    The oven door is closed. ; a painting; a sink, which is turned off. In the
    sink is: nothing.; a substance called soap; a stopwatch, which is
    deactivated. ; a stove, which is turned off. On the stove is: nothing.; a
    table. On the table is: a glass cup (containing nothing).. You also see: A
    door to the bathroom (that is closed); A door to the hallway (that is open);
     A door to the outside (that is closed)
A: open freezer
G: The freezer is now open.
A: look in freezer
G: Inside the freezer is: nothing
...
```

the values for standard deviation (shown in parentheses next to each score), with many tasks having a standard deviation multiple times larger than the ones in Table 1 and included at the bottom of Table 5.

All four models' performance varies a lot with the task. A strong Pearson correlation between the per-task results of all four models (Table 4) hints that some tasks are genuinely more difficult for our GPT-J models.

|  | All train | All train Markov | No variations | Up to 18 games |
|---|---|---|---|---|
| All train | 1 | | | |
| All train Markov | 0.83 | 1 | | |
| No variations | 0.99 | 0.84 | 1 | |
| Up to 18 games | 0.89 | 0.73 | 0.91 | 1 |

Table 4: Pearson correlation between the results in the corresponding columns of Table 5.

# E  Environment step limits

Evaluating the performance of players in a turn-based game raises the question of after how many steps should one report and compare performance. Wang et al. (2022) and Lin et al. (2023) report the performance of models allowed to play up to 100 actions. This paper also reports performance up to a maximum of 100 actions.

However, the training games that ScienceWorld generates suggest that evaluations with more than 100 steps might be necessary. Figure 2 shows that, while the average length of training games is 57 actions and the great majority of games in the training set can be completed in less than 100 steps, a considerable number of games require more than 100 actions to complete. (Lin et al., 2023) cited computational costs as an argument for evaluating up to only 100 actions

Computational costs are less important for the GPT-J models we evaluate since the models are small enough to run on a modern workstation. However, we still report results after a maximum of 100 actions because (1) it makes our results easier to compare to prior literature and (2) because we see diminishing returns from running evaluation past 100 actions. Figure 3 shows our models' ScienceWorld *test* score as a function of the number of game turns/actions. While the Markov assumption GPT-J (All train Markov) flat-lines after about 50

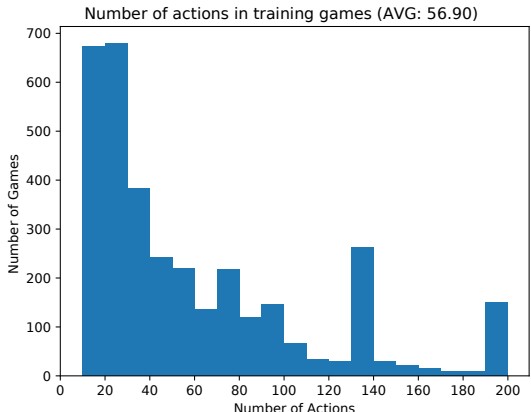

Figure 2: Histogram of the number of actions in games from the *No variations* train set.

turns/actions, the other GPT-J-based models continue to meaningfully accumulate points up to the evaluation limit of 100 turns/actions. Past the 100 actions limit, the increase in ScienceWorld scores is marginal for all our models.

| Task | Games | All train | All train Markov | No variations | Up to 18 games |
|---|---|---|---|---|---|
| **Matter** | | | | | |
| Changes of State (Boiling) | 9 | 1.13 (2.29) | 0.18 (0.10) | 2.51 (3.77) | 7.20 (6.93) |
| Changes of State (Melting) | 9 | 0.31 (0.37) | 0.13 (0.12) | 2.44 (3.74) | 14.42 (11.18) |
| Changes of State (Freezing) | 9 | 0.00 (0.00) | 0.00 (0.00) | 0.96 (2.14) | 8.84 (8.63) |
| Changes of State (Any) | 9 | 0.24 (0.18) | 0.04 (0.06) | 1.91 (3.47) | 19.00 (14.80) |
| **Measurement** | | | | | |
| Use Thermometer | 135 | 63.39 (15.55) | 20.23 (7.18) | 73.59 (21.92) | 29.30 (6.47) |
| Measuring Boiling Point (known) | 109 | 17.76 (8.27) | 13.90 (11.68) | 11.28 (10.98) | 22.88 (13.18) |
| Measuring Boiling Point (unknown) | 75 | 52.03 (17.23) | 53.58 (32.09) | 59.19 (7.20) | 52.60 (21.57) |
| **Electricity** | | | | | |
| Create a circuit | 5 | 82.08 (11.26) | 25.00 (9.10) | 92.36 (4.80) | 77.80 (8.27) |
| Renewable vs Non-renewable Energy | 5 | 59.36 (15.15) | 12.72 (4.20) | 58.72 (11.50) | 45.28 (11.29) |
| Test Conductivity (known) | 225 | 40.02 (21.31) | 16.35 (3.90) | 42.58 (16.30) | 30.57 (14.14) |
| Test Conductivity (unknown) | 150 | 52.20 (2.56) | 37.66 (0.34) | 60.67 (21.67) | 46.15 (3.74) |
| **Classification** | | | | | |
| Find a living thing | 75 | 96.62 (7.55) | 50.10 (4.61) | 99.84 (0.35) | 72.18 (12.33) |
| Find a non-living thing | 75 | 100.00 (0.00) | 66.91 (5.61) | 99.73 (0.60) | 52.68 (19.29) |
| Find a plant | 75 | 99.47 (1.19) | 43.71 (6.29) | 99.40 (0.68) | 46.65 (11.72) |
| Find an animal | 75 | 96.87 (7.01) | 48.68 (6.54) | 99.91 (0.20) | 71.79 (26.36) |
| **Biology** | | | | | |
| Grow a plant | 33 | 11.92 (0.45) | 5.95 (0.14) | 12.47 (0.99) | 9.09 (1.47) |
| Grow a fruit | 33 | 46.72 (4.93) | 11.60 (0.92) | 44.78 (9.55) | 25.08 (11.05) |
| **Chemistry** | | | | | |
| Mixing (generic) | 8 | 41.13 (8.86) | 10.13 (3.38) | 35.63 (8.84) | 28.45 (5.23) |
| Mixing paints (secondary colours) | 9 | 0.00 (0.00) | 10.67 (3.65) | 0.00 (0.00) | 0.67 (1.49) |
| Mixing paints (tertiary colours) | 9 | 0.00 (0.00) | 5.27 (1.88) | 0.00 (0.00) | 0.58 (1.29) |
| **Biology** | | | | | |
| Identify longest-lived animal | 32 | 100.00 (0.00) | 71.25 (2.61) | 98.75 (1.71) | 60.63 (22.30) |
| Identify shortest-lived animal | 32 | 100.00 (0.00) | 66.25 (4.76) | 98.13 (4.19) | 48.13 (17.48) |
| Identify longest-then-shortest-lived animal | 32 | 100.00 (0.00) | 58.75 (7.36) | 98.75 (1.71) | 50.41 (21.90) |
| Identify life stages (plant) | 5 | 45.72 (6.58) | 19.40 (6.47) | 46.56 (19.76) | 29.20 (10.53) |
| Identify life stages (animal) | 4 | 6.00 (10.84) | 8.60 (2.61) | 12.00 (11.51) | 7.00 (5.70) |
| **Forces** | | | | | |
| Inclined Planes (determine angle) | 42 | 76.67 (6.84) | 12.48 (1.05) | 59.83 (8.07) | 39.86 (4.55) |
| Friction (known surfaces) | 348 | 81.86 (4.19) | 12.46 (0.61) | 82.18 (4.73) | 43.92 (2.00) |
| Friction (unknown surfaces) | 42 | 74.12 (8.15) | 13.48 (1.02) | 73.52 (8.23) | 42.00 (6.72) |
| **Biology** | | | | | |
| Mendelian Genetics (known plants) | 30 | 35.79 (17.24) | 8.57 (0.00) | 24.64 (15.14) | 22.85 (6.16) |
| Mendelian Genetics (unknown plants) | 120 | 36.49 (12.83) | 7.59 (0.02) | 29.04 (7.00) | 23.44 (6.34) |
| Overall | 1 819 | 62.57 (4.32) | 24.74 (1.05) | 63.35 (6.94) | 39.78 (2.35) |

Table 5: Results by task for GPT-J-based agents

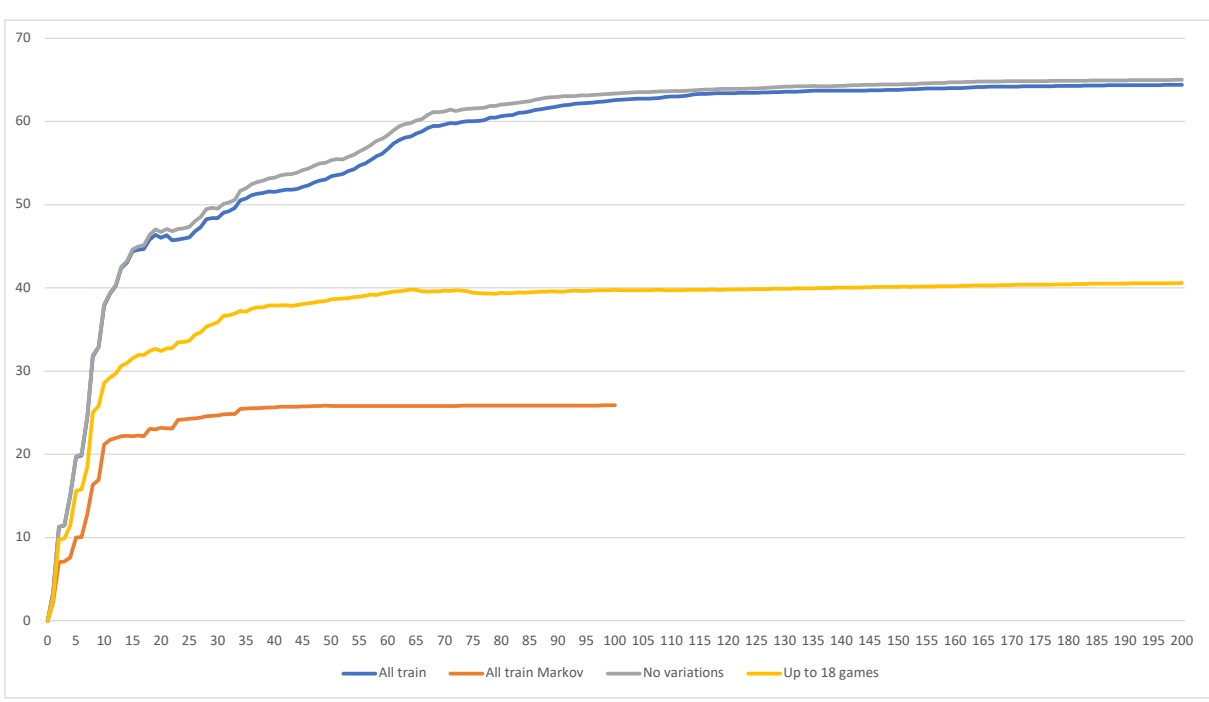

Figure 3: Mean ScienceWorld score as a function of game actions (steps). Each line is the average of evaluating five runs trained with different seeds. We stopped running All train Markov early due to its consistent flat line.