# OpenReview forum: "Remember what you did so you know what to do next"
_EMNLP/2023/Conference — EMNLP 2023 Findings_

### Official Review · Reviewer_bA1q · 2023-08-04

**Typos Grammar Style And Presentation Improvements:** L60-61
**Soundness:** 3

**Excitement:**

4: Strong: This paper deepens the understanding of some phenomenon or lowers the barriers to an existing research direction.

**Paper Topic And Main Contributions:**

This short paper studies the effectiveness of a pre-trained LLM (GPT-J 6B) to tackle ScienceWorld, a simulated game benchmark for elementary science experiments. The empirical results obtained in this paper suggest that an LLM-based transfer learning scheme, when formulated right, can significantly outperform (by a factor of 1.4x) previously built RL-based agents. Importantly, this finding contrasts findings in previous publications which showed LLM could hardly fit this task. Interestingly, the LLM used in this paper (GPT-J 6B) is smaller in model size compared to the LLM used in the previous work that performed poorly (T5 11B), suggesting that diverse pre-training, training task formulation and input context length, three of the biggest differences between the GPT-J and T-5 used for comparison, could be important to make LLM transfer learning work.

**Questions For The Authors:**

Q1: For Table 1, line 1, it is better to include the training set size for DRRN too, so that readers can see a direct comparison of if data size influences performance.

**Reasons To Accept:**

S1: Simple, but convincing and important message. The revelation that LLM transfer learning, when formulated right, can perform well on text-based science games can provide guidance for future work, especially under the circumstance that previous work suggested otherwise.

S2: The method itself (LLM transfer learning with game trajectory converted to text) is simple, which is important: simple methods can usually enjoy wider adaptation.

S3: Straightforward experiments and good analysis support the claims well.

**Reasons To Reject:**

W1: Since the pertaining data used for GPT-J and T-5 is different  (GPT-J uses The Pile and T5 uses C4 + Macaw QA), it is hard to tell how important of a role this difference in pertaining data contribute the final result difference. There should be some experiments controlling this variable - for example, what if the training task formulation and increased context length are used on T5? Would that version of T5 also give us comparable performance similar to the one from GPT-J?

**Reproducibility:**

4: Could mostly reproduce the results, but there may be some variation because of sample variance or minor variations in their interpretation of the protocol or method.

**Reviewer Confidence:**

4: Quite sure. I tried to check the important points carefully. It's unlikely, though conceivable, that I missed something that should affect my ratings.

---

> ### Author Rebuttal · Authors · 2023-08-28
>
> **The reviewer wrote:**
>
> > W1: Since the pertaining data used for GPT-J and T-5 is different (GPT-J uses The Pile and T5 uses C4 + Macaw QA), it is hard to tell how important of a role this difference in pertaining data contribute the final result difference. There should be some experiments controlling this variable - for example, what if the training task formulation and increased context length are used on T5? Would that version of T5 also give us comparable performance similar to the one from GPT-J?”
>
>
> **Reply:**
>
> We agree that controlling for the pretraining data of GPT-J and T5 could provide insights. However, there are factors beyond the training sources. For instance, T5 has an input buffer length of 512 word pieces while GPT-J has an input buffer length of 2048 word pieces. It is, therefore, not possible to train a T5 model with the same historical context size that we used for GPT-J.
>
> _________________
>
> **The reviewer asked**:
>
> > Q1: For Table 1, line 1, it is better to include the training set size for DRRN too, so that readers can see a direct comparison of if data size influences performance.”
>
> **Reply:**
>
> We agree that it would be helpful to illustrate DRRN’s training size, but we have not found a way to express the comparison fairly. DRRN trains with online Reinforcement Learning, meaning that, from the first moment of training, DRRN interacts live with ScienceWorld. For DRRN, we reproduced Ruoyao et al. (2022) training, a process we described in detail in Appendix C of our paper. Ruoyao et al. (2022) trained DRRN with 8 parallel threads each going up to 100k steps in the live ScienceWorld environment. The SW environment was reset after success/failure or after reaching 100 steps. It is difficult to express this amount of training in “games” as DRRN does not observe gold games but learns from its own mistakes. Its early games are only a few actions long, while its latter games are longer. It would be misleading to readers if we estimated DRRN’s training data 8 * 100000 / 100 = 8000 games since this would underestimate the amount of games DRRN plays (remember that early games are shorter due to DRRN’s mistakes).
>
> _________________
>
> Thank you for catching the grammar issue on L60. We will correct it.
>
> _________________
>
> ## References
>
> Wang, Ruoyao, Peter Jansen, Marc-Alexandre Côté, and Prithviraj Ammanabrolu. 2022. “ScienceWorld: Is Your Agent Smarter than a 5th Grader?” In Proceedings of the 2022 Conference on Empirical Methods in Natural Language Processing, 11279–98. Abu Dhabi, United Arab Emirates: Association for Computational Linguistics. https://aclanthology.org/2022.emnlp-main.775.
>
> Lili Chen, Kevin Lu, Aravind Rajeswaran, Kimin Lee, Aditya Grover, Michael Laskin, Pieter Abbeel, Ar- avind Srinivas, and Igor Mordatch. 2021. Decision transformer: Reinforcement learning via sequence modeling. arXiv:2106.01345 [cs].

---

### Official Review · Reviewer_4kDw · 2023-08-05

**Soundness:** 4

**Excitement:**

3: Ambivalent: It has merits (e.g., it reports state-of-the-art results, the idea is nice), but there are key weaknesses (e.g., it describes incremental work), and it can significantly benefit from another round of revision. However, I won't object to accepting it if my co-reviewers champion it.

**Paper Topic And Main Contributions:**

This paper fine-tuned GPT-J for ScienceWorld, and demonstrated a 3.3x improvement over DRRN, a RL-based method. Furthermore, this paper presented a detailed analysis of how various amounts of training data, context length, etc. affected the performance of the model, including a breakdown of what types of errors were most common.

**Questions For The Authors:**

1. Are there ways to disentangle the effect of GPT-J’s pre-trained knowledge from its reasoning ability? Perhaps by exploring how well a non-pre-trained GPT-J performs on these tasks, when simply fine-tuned to perform them?


**Reasons To Accept:**

1. The approach introduced in this paper works empirically well, outperforming an RL-based approach by 3.3x. Moreover, it’s clear this is largely due to increasing context length and conditioning on the full action sequence rather than just the previous action.
2. Perhaps the standout strength of this paper is in its analysis: this paper presented a detailed evaluation of GPT-J in multiple conditions (varying the amount and variation in the training data, amount of context seen by the language model, and presenting a detailed breakdown of error types). This analysis gives solid insight into where benefits come from: in this case likely the additional context and world knowledge in the pre-trained language model.


**Reasons To Reject:**

1. Acknowledging space limitations, some more analysis into why the LM likes to hallucinate invalid objects (even in the full-context settings), given that’s such a massive proportion of the errors, would be useful. This being the predominant source of error seems counter-intuitive to me because the LM should technically be able to refer to its context as to what objects are present, and be able to easily learn not to generate objects that weren’t previously referred to in context.
2. The way things are presented in the paper, it’s still a little bit unclear what type of generalization each condition tests for. Esp. for the “no variation” and “up to 18 games” conditions – these could be articulated a bit clearer either through a better name or a better explanation somewhere. E.g. L156-158 tells us at a very low level that the “no variation” condition is created by including only one unique gold path per task variation. However the implication of this condition is not clearly articulated: why is this interesting to explore? (e.g. is this correct?: “no variation” tests for the ability of generalize within task variants while “up to 18 games” tests for the ability to generalize across task variants). This will make it clearer the implications of the results as to what types of generalization LMs succeed and fail at.
3. Looking at the types of generalization tested, it seems there is one condition that is still missing: that is to say, generalizing /across task classes/. For example, during training LMs do not see any melting tasks, but must do melting tasks during dev and test. I imagine this would be a very hard split and it would be interesting to see whether modern LMs succeed at this already, or whether there is room for improvement.

**Reproducibility:**

4: Could mostly reproduce the results, but there may be some variation because of sample variance or minor variations in their interpretation of the protocol or method.

**Reviewer Confidence:**

4: Quite sure. I tried to check the important points carefully. It's unlikely, though conceivable, that I missed something that should affect my ratings.

---

> ### Author Rebuttal · Authors · 2023-08-28
>
> **The reviewer asked**:
>
> > Are there ways to disentangle the effect of GPT-J’s pre-trained knowledge from its reasoning ability? Perhaps by exploring how well a non-pre-trained GPT-J performs on these tasks, when simply fine-tuned to perform them?
>
> **Reply:**
>
> Excellent question. Yes. One way could be to take an out-of-the-box language model and only use in-context learning to inform the LM how to play the game and what actions are possible. Unfortunately, GPT-J does not have a good enough in-context learning ability to play ScienceWorld. This was one of the first experiments we tried. GPT-J could not learn the ScienceWorld action syntax. We tried this experiment with GPT-3, the first LM that exhibited good in-context learning abilities. While GPT-3 learned the syntax of ScienceWorld commands, it did not play more than a handful of actions before losing the game by issuing an incorrect command or getting stuck in a loop.
>
> The ScienceWorld game as originally published by Ruoyao et al. (2022) incorporates 4 pairs of known vs. unknown tasks designed to probe the difference between knowledge the system has and knowledge it learns through game play. In the known condition, the object is assigned a real-world name (e.g., *metal fork*) and players can rely on prior knowledge about that object. In the unknown condition, the object is an English language variable name (e.g., *unknown substance B*) forcing players to perform experiments and apply what they learn to complete the task. Table 5 in the appendix presents results broken by task, thereby allowing readers to see each of the four comparisons directly (e.g., comparing *Measuring Boiling Point (Known)* with *Measuring Boiling Point (unknown)*. At least at the task level, the results here are difficult to interpret. For two of the pairings, results on unknown objects are much higher in absolute for all four of the conditions we explore. For the other two tasks, there is less of a pattern. In some training conditions, known is higher than unknown; in others, the scores vary in absolute value. There are possible explanations, for example: (1) prior knowledge about an object could be harder for the model to overcome in the case of unexpected situations (e.g., encountering a broken stove); or (2) similarity in the naming of the unknown objects could simplify the transformer’s ability to transfer information across tasks, e.g., making it easier to share knowledge between *Unknown Substance 1* and *Unknown Substance 2* than it is to transfer across, e.g., *Metal Fork* and *Glass Jar*.
>
> Given the scope of a short paper, we have not performed sufficient analysis to explore why, but if accepted we appreciate the insight that this would be good fodder to include as a part of future questions raised by this work.
>
> _________________
>
> **The reviewer wrote**
>
> > “Acknowledging space limitations, some more analysis into why the LM likes to hallucinate invalid objects (even in the full-context settings), given that’s such a massive proportion of the errors, would be useful. This being the predominant source of error seems counter-intuitive to me because the LM should technically be able to refer to its context as to what objects are present, and be able to easily learn not to generate objects that weren’t previously referred to in context.”
>
> **Reply**:
>
> In the full-context setting, 4.28% of emitted actions include invalid objects (row 3 in Table 1). This percentage increases as the training data decreases. It also increases drastically for the *All Train Markov* GPT-J. We agree that hallucinations are interesting, but we find it difficult to properly address them in a short 4-page paper alongside our other analyses. Please note that even far larger models like GPT-3 still hallucinate entities and whole facts.
>
> _________________
>
> **The reviewer wrote**:
>
> > The way things are presented in the paper, it’s still a little bit unclear what type of generalization each condition tests for. Esp. for the “no variation” and “up to 18 games” conditions – these could be articulated a bit clearer either through a better name or a better explanation somewhere. E.g. L156-158 tells us at a very low level that the “no variation” condition is created by including only one unique gold path per task variation. However the implication of this condition is not clearly articulated: why is this interesting to explore? (e.g. is this correct?: “no variation” tests for the ability of generalize within task variants while “up to 18 games” tests for the ability to generalize across task variants). This will make it clearer the implications of the results as to what types of generalization LMs succeed and fail at.
>
> **Reply**:
>
> We agree that the paper needs a better explanation of the three evaluation settings, including a better name for the settings. We will revise the explanation. In the meantime, let us try to explain what each condition tests for.
>
> The *All train* set aims to answer the question: *What is the highest SW Score GPT-J can achieve if it has access to a large set of training data?* The *All train* set contains 7359 games covering the 30 classes of tasks contained in ScienceWorld and listed in Table 5. Each task category is represented by several specific task instantiations contained in ScienceWorld. For example, if the category is *Changes of State (Melting)*, some actual instantiations are: boil water, boil water but the stove is broken, boil water but the sink is broken, boil chocolate, boil lead, etc. For each instantiation, we collect from ScienceWorld up to three solution variations, i.e., unique sequences of actions that solve the task.
>
> The *No variations* set aims to answer the question: *How much does GPT-J’s performance drop if it does not have access to a large amount of training data?* The *All train* set frequently includes more than one correct sequence of actions to achieve the goal. If LMs learn a lot during pretraining, they shouldn’t require that much training to play a game that tests for knowledge that should have been gained in pre-training. *No variations* covers the same tasks as *All train*, the same instantiations as *All train*, but it only includes a single gold path per task instantiation (i.e., no variations).
>
> The final training set, *Up to 18 games* acknowledges that in the prior two train sets, some tasks are overrepresented and tries to answer: *Can GPT-J learn to play SW from only 480 games but which are balanced over the tasks?* To get an idea of the imbalance in the *No variations* training set, multiply the *Games* column in Table 5 by 2 to get the number of games for each task in the *No variations* set. For example, *No variations* contains 250 games where the player must measure the conductivity of a known material, but only 18 games where the player must mix paints to obtain tertiary colors.
>
> Even with a training set of only 480 games, GPT-J still outperforms the RL system (DRRN). Interestingly, Table 5 in our paper shows that for some underrepresented tasks *Up to 18 games*-trained GPT-J improves its performance (e.g., Changes of state), but for many others, it does not (e.g., Mixing paints).
>
> _________________
>
> **The reviewer wrote**:
>
> > Looking at the types of generalization tested, it seems there is one condition that is still missing: that is to say, generalizing /across task classes/. For example, during training LMs do not see any melting tasks, but must do melting tasks during dev and test. I imagine this would be a very hard split and it would be interesting to see whether modern LMs succeed at this already, or whether there is room for improvement.
>
> **Reply**:
>
> We have tested training on most of the classes of tasks and tested on classes of tasks that were not seen during training. We did not include those results in the draft due to the four-page limitation. In short, for some classes of tasks, performance is poor for some classes of task; for other new classes of tasks, performance is good.  If this short paper were accepted, we could use the extra page to report those results or would gladly add them to the appendix.
>
> _________________
>
> ## References
>
> Wang, Ruoyao, Peter Jansen, Marc-Alexandre Côté, and Prithviraj Ammanabrolu. 2022. “ScienceWorld: Is Your Agent Smarter than a 5th Grader?” In Proceedings of the 2022 Conference on Empirical Methods in Natural Language Processing, 11279–98. Abu Dhabi, United Arab Emirates: Association for Computational Linguistics. https://aclanthology.org/2022.emnlp-main.775.

---

### Official Review · Reviewer_4gZk · 2023-08-09

**Soundness:** 3

**Ethical Concerns:**

Yes

**Excitement:**

3: Ambivalent: It has merits (e.g., it reports state-of-the-art results, the idea is nice), but there are key weaknesses (e.g., it describes incremental work), and it can significantly benefit from another round of revision. However, I won't object to accepting it if my co-reviewers champion it.

**Missing References:**

There are several works that use LMs to interact with text-based environments. Here are a few examples:
- https://arxiv.org/abs/2107.08408
- https://aclanthology.org/2022.emnlp-main.86/

**Paper Topic And Main Contributions:**

The paper finetunes GPT-J on all 30 classes of tasks of the ScienceWorld environment using 4 different methods:

1. **All train Markov** uses the entire training data but only provide the prior action and game observation as input (Markov assumption).
2.  **All train** uses the entire training set but "fill's GPT-J's input buffer" by previous actions and observation.
3. **No variations** Like (2) but using half of the training data.
4. **Up to 18 games** Like (2) but trained on a small subset of 18 task variations per task.

The model interacts with the environment through a dialog where the agent (GPT-J) issues actions and the game replies with the current observation and score. The authors also use a *preconditions checker* to assist the LLM. It intercepts GPT-J's output in case the model is providing redundant actions to the environment (opening a door that is already open).

The authors show the training using method #2 triples the average score compared to method #1, while method #4 only doubles the result as compared to previous SoTA, and that the preconditions system is unnecessary if all of the training data is used.

**Reasons To Accept:**

- The paper shows that LLMs can achieve decent results on the ScienceWorld environment by making use of its world and commonsense knowledge.

**Reasons To Reject:**

There are many reasons to reject this paper:
- The paper is poorly written and structured. It contains words such as "explore *stuffing* the LLM’s input buffer" and "It is *seductive* to look at our experience with GPT-J" that are not suitable for an academic paper.
- The paper's contribution is minimal. It's trivial to see that including more context for GPT-J will improve the result compared to only giving it the previous action and observation, and including less training data will reduce that result.
- The experiments design are not clearly motivated. Specifically, the **no variations** and the **Up to 18 games** finetuning methods.
- The authors do not test their model on novel unseen task, just a variation of a task seen during training.

In general, I think the paper requires serious revisions and would then might be considered a good workshop submission, but not a conference submission.

**Reproducibility:**

4: Could mostly reproduce the results, but there may be some variation because of sample variance or minor variations in their interpretation of the protocol or method.

**Reviewer Confidence:**

4: Quite sure. I tried to check the important points carefully. It's unlikely, though conceivable, that I missed something that should affect my ratings.

**Typos Grammar Style And Presentation Improvements:**

There are a lot of improvements that can be made:

1. Tables 1 and 2 needs proper styling. Please see for reference: https://people.inf.ethz.ch/markusp/teaching/guides/guide-tables.pdf
2. Please avoid using words such as "stuffing" and "seductive"
3. The Limitations and Ethics sections need to be rewritten entirely. Saying that your finetuned LLM shouldn't interact in the real world is a trivial point. For example, the paragraph below shouldn't be there.

> LLMs' unpredictable behavior, such an approach could result in material damage or even injure humans, animals, or the environment. This warning also applies to cases where LLMs are allowed to operate computer APIs which, despite their virtual nature, can have undesired effects in the real world. Should LLMs be allowed to operate APIs or real-world machinery, they should not be given complete control over the tools but operate within carefully chosen boundaries strictly enforced through software or physical means.

4. Saying that you are based in California and that the:
> California grid was 59% powered by energy from zero-carbon sources

should not be an Ethics statement.

---

> ### Author Rebuttal · Authors · 2023-08-28
>
> First, we particularly appreciate those changes that are specific. On some other points, there seems to be a difference of opinion with us and possibly with the other reviewers; we address those first.
>
> * You made the following point:
>
> > It's trivial to see that including more context for GPT-J will improve the result compared to only giving it the previous action and observation, and including less training data will reduce that result.
>
> Those two points may be obvious to some. However, the context for our study appeared in EMNLP 2022, Ruoyao et al. [1]. The authors claim,
>
> > "We evaluate the performance of 5 state-of-the- art reinforcement learning and language model agents on these benchmarks, empirically demonstrating that existing agents for text-based games as well as novel models adapted from transformer-based scientific question-answering solvers perform poorly on tasks (such as melting ice) that 5th grade science students can perform with ease."
>
> We feel that negating the published claim that transformers perform poorly is worth noting and worth showing that providing adequate context to the transformer is what the original authors missed.
>
> _____________
>
> * You judge that "The paper's contribution is minimal."  Below is an extract from the ACL’s guidelines for short papers
>
> > "Please note that a short paper is not a shortened long paper. Instead short papers should have a point that can be made in a few pages. Some kinds of short papers are:
> >
> >        ◦ "A small, focused contribution
> >        ◦ "A negative result
> >        ◦ "An opinion piece
> >        ◦ "An interesting application nugget"
>
> We feel that correcting incorrect claims in a recent paper, showing remarkably improved (>3x) performance on the 30 classes of tasks in ScienceWorld, estimating confidence intervals on measured performance, and showing that far less training data is needed than employed in the earlier paper while simultaneously doubling scores with ~7% of the training data, together qualify as a small, focused contribution.
>
> _____________
>
> On the following points, we are glad to make concrete changes:
>
> * We agree when you point out that "The authors do not test their model on novel unseen task, just a variation of a task seen during training." In fact, we have tested training on most of the classes of tasks and tested on classes of tasks that were not seen during training. We did not include those results in the draft due to the four-page limitation. In short, for some classes of tasks, performance is poor for some classes of task; for other new classes of tasks, performance is good.  If this short paper were accepted, we could use the extra page to report those results or would gladly add them to the appendix.
> * You state that "explore stuffing the LLM’s input buffer" and "It is seductive to look at our experience with GPT-J" are not suitable for an academic paper and later mention stuffing and seductive again.  We will replace *stuffing* with *filling* and *seductive* with *attractive*.
> * We will add the two references you mention to the discussion of related work and distinguish our work from theirs:  https://arxiv.org/abs/2107.08408 and https://aclanthology.org/2022.emnlp-main.86/
> * We will adopt the table style you recommend in https://people.inf.ethz.ch/markusp/teaching/guides/guide-tables.pdf for our tables 1 and 2, though those guidelines do not seem to be specified by the ACL.
> * We can make the deletions from the Ethics Statement per your request. However, please note the ACL’s guidance on the Ethics Statement; The EMNLP 2023 call for papers [2] points authors to the EMNLP 2021 FAQ [3], where authors are urged to address misuse potential and the environmental impact of computing used for experiments. Thus, what you want us to delete seemed pertinent given the ACL’s FAQ.
> * You scored the reproducibility of our paper using the second lowest possible grade. We will amend the draft to state where the original ScienceWorld training data may be obtained:
>     * The training and evaluation data we used are generated by ScienceWorld (Ruoyao et al., [1]), an Apache-licensed open-source environment. To improve the reproducibility of our work beyond the details in the paper, we will include a link for readers to download the exact training data sets for all evaluations.
>     * The LM model we fine-tune is GPT-J, an Apache-licensed open-source model, trained on The Pile, an open-source corpus.
>     * We followed the EMNLP guidelines on Optional Supplementary Materials [4] and put in the appendix sample inputs (Appendix A), hyperparameter details (Appendix B), and evaluation setup (Appendix C). All these are also pointed to from the paper body (L174, L195, L219).
>
> ## References
>
> [1] Wang, Ruoyao, Peter Jansen, Marc-Alexandre Côté, and Prithviraj Ammanabrolu. 2022. "ScienceWorld: Is Your Agent Smarter than a 5th Grader?" In Proceedings of the 2022 Conference on Empirical Methods in Natural Language Processing, 11279–98. Abu Dhabi, United Arab Emirates: Association for Computational Linguistics. https://aclanthology.org/2022.emnlp-main.775
>
> [2] https://2023.emnlp.org/calls/main_conference_papers/#ethics-policy
>
> [3] https://2021.emnlp.org/call-for-papers/ethics-faq
>
> [4] https://2023.emnlp.org/calls/main_conference_papers/#optional-supplementary-materials

---

### Meta-Review · Area_Chair_QW1m · 2023-09-19

**Recommendation:** 3

**Metareview:**

This short paper offers a focused exploration of the effectiveness of LLMs, specifically GPT-J 6B, when applied to the ScienceWorld domain. The study involves fine-tuning GPT-J for ScienceWorld and highlights an 3.3x enhancement in performance compared to DRRN, which is an RL-based method.
Additionally, the paper presents various analyses of factors influencing performance, including the effects of training data volume and context length.  It also offers a breakdown of the most common error types encountered during experimentation. However, two reviewers find the paper not exciting enough. Therefore, I recommend accepting the paper to findings.

---

### Decision · Program_Chairs · 2023-10-07

**Decision:**

Accept-Findings

**Comment:**

This short paper offers a focused exploration of the effectiveness of LLMs, specifically GPT-J 6B, when applied to the ScienceWorld domain. The study involves fine-tuning GPT-J for ScienceWorld and highlights an 3.3x enhancement in performance compared to DRRN, which is an RL-based method.
Additionally, the paper presents various analyses of factors influencing performance, including the effects of training data volume and context length.  It also offers a breakdown of the most common error types encountered during experimentation. However, two reviewers find the paper not exciting enough. Therefore, I recommend accepting the paper to findings.